# The Safety Profile of General and Local Anaesthetic Agents: Data Collected during 20 Years of Spontaneous Reporting Activities in the Campania Region (Southern Italy)

**DOI:** 10.3390/ph14121261

**Published:** 2021-12-03

**Authors:** Francesca Gargano, Cristina Scavone, Gabriella di Mauro, Alberto Della Corte, Alice Zoccoli, Francesco Rossi, Giovanni Francesco Nicoletti, Annalisa Capuano

**Affiliations:** 1UOC Anestesia e Rianimazione, Policlinico Universitario Campus Bio-Medico, 00128 Rome, Italy; f.gargano@unicampus.it; 2Campania Regional Centre for Pharmacovigilance and Pharmacoepidemiology, 80138 Naples, Italy; gabriella.dimauro@unicampania.it (G.d.M.); francesco.rossi@unicampania.it (F.R.); annalisa.capuano@unicampania.it (A.C.); 3Department of Experimental Medicine, University of Campania “Luigi Vanvitelli”, 80138 Naples, Italy; albertodellacorte@live.it; 4Clinical Innovation Office, Università Campus Bio-Medico, 00128 Rome, Italy; a.zoccoli@unicampus.it; 5Multidisciplinary Department of Medical Surgical and Dental Sciences, Plastic Surgery Unit, University of Campania “Luigi Vanvitelli”, 80138 Napoli, Italy; giovannifrancesco.nicoletti@unicampania.it

**Keywords:** general anaesthetics, local anaesthetics, safety, spontaneous reporting system

## Abstract

Background: General and local anaesthetics are widely used during surgery. These drugs have peculiar safety profiles, being commonly associated with mild and reversible local adverse drug reactions (ADRs), but also with more severe and systemic ADRs, including respiratory and cardiovascular depression and anaphylaxis. Methods and Objectives: We carried out a descriptive analysis of Individual Case Safety Reports (ICSRs) sent to the Campania Regional Centre of Pharmacovigilance (Southern Italy) from 2001 to 2021 that reported general or local anaesthetics as suspected drugs, with the aim of describing their overall characteristics, focussing on the ADRs’ seriousness and distribution by System Organ Class (SOC) and Preferred Term (PT). Results: A total of 110 ICSRs documenting general or local anaesthetics were sent to the Italian pharmacovigilance database during 20 years of spontaneous reporting activities in the Campania region. ADRs mainly occurred in patients with a median age of 48 years and in a slightly higher percentage of men. ADRs were more commonly classified as not serious and had a favourable outcome. In terms of ADRs’ distribution by SOC and PT, both general and local anaesthetics were associated with general and cutaneous disorders, with common ADRs that included lack of efficacy, rash, and erythema. In addition, general anaesthetics were associated with the occurrence of respiratory ADRs, while local anaesthetics were associated with the occurrence of nervous ADRs. Conclusion: Even though a limited number of ICSRs documenting anaesthetics-induced ADRs were retrieved from the Italian spontaneous reporting database in the Campania region, we believe that the continuous monitoring of these drugs is highly recommended, especially among the frail population.

## 1. Introduction

Anaesthesia can be successfully achieved with a wide range of drugs that include general and local anaesthetics, but also analgesics, sedatives, and neuromuscular blocking drugs. General anaesthetics represent one the most important drugs for surgical procedures. They induce a state of controlled and reversible loss of consciousness that is the result of sedation, amnesia, and muscle paralysis [1]. Drugs used in general anaesthesia belong to different therapeutic classes with diverse mechanisms of action (mainly an enhancement of inhibitory neurotransmissions or a reduction of excitatory neurotransmissions), and are generally administered via different routes. Among intravenous anaesthetics, propofol is a phenol agent that is used for the induction and maintenance of anaesthesia; etomidate is an ultrashort-acting, non-barbiturate hypnotic that is usually used only for induction; while ketamine is a dissociative anaesthetic that produces intense analgesia. Inhalational anaesthetics (such as nitrous oxide, which is a gas, and sevoflurane, desflurane and isoflurane, which are liquids) are rapidly absorbed at the level of pulmonary alveoli. They are mainly used for the maintenance of anaesthesia. Another drug class that has been widely used for the induction of anaesthesia is represented by barbiturates (mainly thiopental) [2]. However, given their peculiar safety profile (risk of addiction, overdose, and death), the use of barbiturates was gradually replaced by safer anaesthetic agents for intravenous administration [3]. Intravenous sedatives, mainly benzodiazepines, are commonly used as premedication agents. Among this class, midazolam is the most widely used, both before the anaesthesia induction for its sedative, anxiolytic, and amnestic properties, and for co-induction in order to speed up the onset of hypnosis and to reduce the doses required of other anaesthetics [4,5]. Synthetic opioids such as sufentanil, remifentanil, and fentanyl are commonly used for their potent analgesic effect in surgery units where ventilatory support is readily available. Lastly, neuromuscular blocking drugs (such as succinylcholine, cisatracurium, and rocuronium) act on the postsynaptic membrane of nicotinic cholinergic receptors [6,7,8]. On the other hand, the effects of local anaesthetics derive from the blocking of nerve impulse transmission in the peripheral and central nervous system (CNS), even though they do not cause CNS depression or an altered mental status. These drugs are commonly used to induce reversible anaesthetic effects during local surgical procedures (epidural and spinal blocks). Among this class, lidocaine, mepivacaine, ropivacaine, bupivacaine, levobupivacaine, and procaine are the local anaesthetics most commonly used [9,10]. As reported in the latest OsMed Report of the Italian Medicines Agency (AIFA-https://www.aifa.gov.it/documents/20142/1542390/Rapporto-OsMed-2020.pdf, accessed on 26 November 2021), an increase in the use of injecting drugs was recorded in Italy in 2020, including neuromuscular blocking drugs, hypnotics, sedatives, and general anaesthetics.

Regarding to the safety profile of anaesthetics, the occurrence of adverse drug reactions (ADRs) is common during general anaesthesia. Indeed, general anaesthetic can commonly induce confusion or memory loss, dizziness, urinary retention, nausea, vomiting, and chills [11]. In addition, depending on the type of general anaesthetic that is used, further ADRs could commonly occur, such as a profound respiratory depression, hypotension, and bradycardia until cardiac arrest due to the induction dose of propofol; nausea or vomiting after etomidate; laryngospasm and hallucinations after ketamine; or meiosis, respiratory depression, bradycardia, constipation, and urinary retention after opioids [11]. Inhalational anaesthetics can induce cardiovascular, renal, hepatic, and gastrointestinal toxicities [12]. On the other hand, benzodiazepines can be associated with other types of ADRs, including thrombophlebitis, thrombosis, anterograde amnesia, drowsiness, ataxia, falls, and confusion. Some of these ADRs are more common in the elderly [13]. Lastly, depending on the type of neuromuscular blocking drug, cardiac and cutaneous ADRs may occur [14,15]. Regarding the safety profile of local anaesthetic, the most common ADRs are represented by those occurring in the application site, which includes pain, erythema, and oedema [16]. When the plasmatic level of the anaesthetic rises to concentrations above those recommended, so-called local anaesthetic systemic toxicity could occur. This is the most severe, but rare, ADR associated with local anaesthetics, with symptoms that include paraesthesia, dysarthria, diplopia, ear disturbances, seizures, hypertension, and tachycardia. Severe cases progress to CNS and respiratory depression, with cardiovascular effects that include myocardial depression, prolonged conduction interval, bradycardia, and heart failure [17]. An overview of ADRs associated with drugs used during general and local anaesthesia is reported in Table 1.

Considering that general and local anaesthetics are widely used during surgical procedures, and given their peculiar safety profiles, we carried out a pharmacovigilance study using data from the Italian spontaneous reporting system to describe regional Individual Case Safety Reports (ICSRs) documenting the aforementioned suspected drugs, with the aim of describing the overall characteristics of these ICSRs, focussing on their seriousness and distribution by System Organ Class (SOC) and Preferred Term (PT).

## 2. Results

From 1 January 2001 until 7 September 2021, the Campania Regional Centre of Pharmacovigilance received 110 ICSRs, covering 176 ADRs, that reported general and local anaesthetic medicines as suspected (50 ICSRs and 60 ICSRs, 86ADRs and 90 ADRs, respectively). Overall, anaesthetics-induced ADRs occurred in patients with a median age of 48.5 years and in a slightly higher percentage of male patients (51%) than female patients (46.3%). The majority of ICSRs (58.2%) reported ADRs that were classified as not serious; however, the remaining ICSRs mainly reported serious ADRs that were clinically relevant or life-threatening (15.5% and 12.7%, respectively) (Table 2). Regarding ICSRs which reported life-threatening ADRs, 14 cases were retrieved from the National Pharmacovigilance Network (Rete Nazionale di Farmacovigilanza—RNF) in the Campania region. These mainly reported general anaesthetics as suspected (in 11/14; propofol was reported as suspected in 8 ICSRs). The age of the patients ranged from 2 to 67 years; half of these ICSRs referred to female patients and half to males. In most of cases, life-threatening ADRs were represented by hypersensitivity reactions that occurred shortly after the administration of the anaesthetic, including many cases of bronchospasm, rash, and cardiovascular symptoms (hypotension, bradycardia, and tachycardia) (Table 3). For all ICSRs (serious and not serious), the outcome was favourable in 71% of cases and unfavourable in 5.4% of cases (Table 2). Lastly, almost 97% of ICSRs (*n* = 107) were sent by healthcare professionals, while the remaining 3% were sent by patients (data not shown).

By observing the ICSRs related to general and local anaesthetics, some differences can be highlighted. For instance, the percentage of not-serious ADRs is higher for local anaesthetic than for general ones (65% vs. 50%, respectively), while the percentage of serious life-threatening ADRs is higher for general anaesthetic than for local ones (22% vs. 5%, respectively) (Table 2).

Lastly, the distribution of ADRs by SOC and PT revealed that for general anaesthetic ADRs were more commonly related to the SOC “skin and subcutaneous tissue disorders” (22%); followed by “respiratory, thoracic, and mediastinal disorders” (14%); and “general disorders and administration site conditions” (13%). Regarding the distribution by PT, we found that for cutaneous disorders the most reported ADRs were rash and erythema (twelve cases and four cases, respectively), bronchospasms for respiratory disorders (five cases), and lack of efficacy for general disorders (ten cases) (Table 4). SOCs that were most commonly reported for local anaesthetic were “general disorders and administration site conditions” (58% of all local anaesthetic-induced ADRs), followed by “skin and subcutaneous tissue disorders” (11%), and “nervous system disorders” (9%). Among these SOCs, the PTs most commonly reported were a lack of efficacy and pain for general disorders (forty-two cases and seven cases, respectively), rash and angioedema for cutaneous ADRs (two cases each), and paralysis for CNS disorders (two cases) (Table 4).

## 3. Discussion

During the first 20 years of spontaneous reporting activities in the Campania region (Southern Italy), 110 ICSRs that documented general and local anaesthetic medicines as suspected were collected. Reported ADRs occurred in patients with a median age of 48 years and in a slightly higher percentage of men (51%) than women (46.3%). To our knowledge, only a few drug utilisation studies evaluated the use of anaesthetics by age groups. As reported by Purdon and Turrentine, a high proportion of the patients who undergo anaesthetic procedures in the USA are aged 60 years or older [18,19]. Despite the increase of surgical procedures for the elderly, a retrospective study that examined the use of anaesthetic drugs in more than 30 000 patients, showed a decreased utilisation of fentanyl, propofol, thiopental, isoflurane, and midazolam with increasing age [20]. Thus, the older age of patients who experienced anaesthetics-induced ADRs in our study could be related both to differences in how patients of different age groups react to anaesthetics and modifications in liver and kidney function in the elderly population. Indeed, the elderly seem to be more sensitive to anaesthetic agents and their potential complications, including ADRs [21]. Regarding the distribution of ICSRs by gender, we found a slightly higher percentage of male patients experiencing anaesthetic-induced ADRs, which seems to be in contrast with the literature data reporting a higher risk of ADRs from the suspected drugs for women due to sex-related factors in the pharmacokinetic and pharmacodynamic behaviour of drugs. Indeed, the risk for women of developing an ADR is 1.5- to 1.7-fold higher compared to men [22,23]. This was previously reported for other classes of drug-induced ADRs in other studies [24,25,26]. However, as reported by Nicolson et al., many drug classes exhibit sex-based variation in pharmaceutical efficacy and toxicity, including anaesthetics [27].

Most of the ICSRs, especially those related to local anaesthetics, reported ADRs that were classified as not serious. In addition, the outcome was favourable in 71% of cases. This is in line with the literature data suggesting that local anaesthetics can be considered safe medications when they are used in proper doses and concentrations [28], even though serious ADRs, both local and systemic, could occur. Indeed, regarding the ICSRs documenting serious ADRs, we found that clinically relevant or life-threatening ADRs were the most common. In particular, life-threatening ADRs were mainly reported for general anaesthetics, especially propofol. In most cases, life-threatening ADRs were represented by hypersensitivity reactions. As reported by Gerald W Volcheck, hypersensitivity reactions, which could occur both during general and local anaesthesia, are associated with significant morbidity and mortality. The occurrence of such ADRs depend on many factors, including the large number of medications administered in the same time frame, such as neuromuscular blocking agents, antibiotics, and latex, which are the most common causes of anaesthesia-associated allergic reactions [29]. The incidence of anaphylaxis during anaesthesia was estimated to be between 1/1250 and 10 000 [30], while the associated mortality rate varies across countries, ranging from 4.76% in Japan, 4% in the United States, 9% in the United Kingdom, and 0 to 1.4% in Western Australia [31,32,33]. Reasons underlying the high mortality rate associated with anaphylaxis include patient frailty (mainly depending on comorbid conditions), the cardiopulmonary depressive effects of anaesthetics, and the intravenous administration of massive doses of general anaesthetics [29].

Almost 97% of ICSRs were sent by healthcare professionals while the remaining 3% were sent by patients. This is in line with findings from previous studies [34,35] and could be related to the new pharmacovigilance legislation which has further increased the involvement of healthcare professionals (HCPs) in pharmacovigilance activities [36,37]. On the other hand, the limited contribution of patients to the collection of ICSRs could be related to several factors, including a low education in terms of pharmacovigilance that still persists, not only among patients/citizens but also in the specific clinical field of our study. Indeed, patients admitted to surgery units might be less inclined to report ADRs compared to patients admitted to other clinical units because of the administered drugs and their related memory loss effects.

Regarding the distribution of ADRs by SOCs and PTs, we found that general anaesthetics were mainly associated with cutaneous and respiratory disorders and that the most commonly reported PTs were rash, erythema, and bronchospasms. In our opinion, these symptoms represent the main features of allergic reactions, which could occur abruptly with symptoms that include cardiovascular collapse, bronchospasm and cyanosis, sinus tachycardia and dysrhythmias, cutaneous and mucosal angioedema, urticaria and generalised erythema, vomiting, and abdominal cramps [38]. Among general disorders, the most commonly reported PT was the lack of efficacy. During general anaesthesia, this event is commonly defined as anaesthesia awareness, indicating the situation that occurs when a patient becomes aware of some or all events occurring during surgery. The risk for awareness is almost 0.1%; young age, female sex, and cardiac or obstetric surgery carries a higher risk for this event [39,40].

On the other hand, we found that local anaesthetics were mainly associated with general disorders (mainly lack of efficacy), cutaneous and nervous disorders (with common signs and symptoms that included rash, angioedema, and paralysis). In our opinion, the occurrence of cutaneous ADRs is not surprising considering that, as previously reported [16], local anaesthetics are mainly associated with ADRs occurring in the application site. In addition, in line with our results, another study carried out in France based on data from the spontaneous reporting system reported that the block failure—defined as insufficient or inadequate sensory anaesthesia—was the most frequent ADR reported with allergic manifestations (oedema, skin eruptions, anaphylactic reactions, and urticaria) and neurological symptoms (including paralysis) [41].

Regarding cases with a lack of efficacy related to both types of medications, we cannot exclude the possibility that they could represent the consequences of a clinical errors in the anaesthetic procedure/administration, rather than an ADR. Indeed, according to Orser et al., medication errors, deriving for instance from misidentification of ampoules, vials, and syringes, still represent a leading cause of ADRs among patients undergoing anaesthesia [42].

Apart from the well-known limitations of spontaneous reporting systems (spontaneous reports represent a voluntary communication from patients and physicians; the underreporting phenomenon; the lack of important clinical data in ICSRs (including gender, ADR seriousness/outcome, and anaesthetic dose) that—in our opinion—can be considered a consequence of the voluntariness of spontaneous reporting of suspected ADRs), which inevitably have affected our study, there are further limitations strictly related to our study that need to be stated. Firstly, we carried out a study using data from a single Italian region, which limits the generalisability of our results. We are aware that further studies based on data from larger pharmacovigilance databases, e.g., the EudraVigilance, are strongly needed in order to confirm the incidences of the anaesthetics-induced ADRs identified in our study. Secondly, many ICSRs did not report information on the outcome of ADRs; in our opinion, this was possibly related to the setting where we carried out the study, considering that at surgery units the patient’s stay is usually limited in time. Thirdly, a reporting bias may have affected the degree of seriousness of the anaesthetics-induced ADRs; indeed, almost all of the ICSRs in our study were reported by healthcare professionals who seem to be more inclined to report serious ADRs.

On the other hand, we were able to provide a descriptive analysis of the spontaneous reporting system activities—related to anaesthetic drugs—that have been carried out in the Campania region since 2001. In addition, as recently reported by Raschi et al. [43], data from pharmacovigilance databases (such as the RNF that we used for our study) represent an unprecedented opportunity to provide a real-time overview of major drugs-related toxicities. As a matter of fact, among anaesthetic-induced ADRs, we described those that were considered as life-threating, highlighting that many cases were related to hypersensitivity reactions, but we also found many cases related to anaesthesia block failure. Given their seriousness, in our opinion, the occurrence of these ADRs should be prevented as much as possible (for instance, through the consideration of the patient’s history of allergies, and providing sufficiently deep anaesthesia or optimising the use of muscle relaxants) and strictly monitored.

## 4. Materials and Methods

### 4.1. Data Sources

The Italian spontaneous reporting system is coordinated by the Italian Medicines Agency (AIFA). The collection of ICSRs is made through the Italian spontaneous reporting database (Rete Nazionale di Farmacovigilanza—RNF) that was established by the AIFA in 2001. The RNF ensures the collection, management, and analysis of ICSRs across the entire Italian territory. Pharmacovigilance -responsible persons from local health units, hospital enterprises, and the National Institutes for Research and Treatment across the Italian regions and autonomous provinces have the credentials to access the RNF and upload the ICSRs made by healthcare professionals or patients/citizens, which are collected from spontaneous reports or reports from observational studies, or from those reporting ADRs that occurred during a compassionate use of drugs. Each Italian region has a regional centre of pharmacovigilance which is responsible for the quality control of ICSRs uploaded at a regional level for causality assessment, the continuous medical education in pharmacovigilance, and, together with the Italian Medicine Agency, for data-mining and signal detection activities.

We retrieved all the ICSRs that reported a drug belonging to the ATC N01A (general anaesthetic) or N01B (local anaesthetic) as suspected from the RNF for one Italian region (Campania region, Southern Italy—Appendix A), from 1 January 2001 to 7 September 2021. C.S. and G.d.M carried out the database interrogation by specifying the suspected drugs (by ATC code), the Italian region, and the time frame. ICSRs from the literature were excluded.

### 4.2. Data Analysis

Information on gender, median age, seriousness and outcome of ADR, suspected anaesthetic drug, type of reporter, and ADR distribution by System Organ Class (SOC) and Preferred Term (PT) was provided for all ICSRs. According to the document “ICH Topic E 2 D Post Approval Safety Data Management” [44], ADRs were categorised as serious when they resulted in death, or were life-threatening, required inpatient hospitalisation or prolongation of existing hospitalisation, resulted in persistent or significant disability or incapacity, or resulted in a congenital anomalies/birth defects or clinically relevant conditions based on clinical judgments. Otherwise, they were classified as not serious. Lastly, according to the national pharmacovigilance rules, the ADRs’ outcome was categorised into three main categories: favourable, when the ADR recovered or improved; unfavourable, when the ADR was resolved with sequelae, remained unchanged, or induced the patient’s death; and not available.

### 4.3. Compliance with Ethical Standards

Safety data derived from the Italian spontaneous reporting system are anonymous and in compliance with the ethical standard. Therefore, no further ethical measures were required.

## 5. Conclusions

We descriptively analysed ICSRs related to general and local anaesthetics that were sent to the RNF from 2001 to 2021. Our results demonstrated that a limited number of ICSRs (*n* = 110) were reported during 20 years of spontaneous reporting activities in the Campania region. Reported ADRs mainly concerned patients with a median age of 48 years and a slightly higher percentage of men. ADRs were more commonly classified as not serious and their outcome was favourable in 71% of cases. Lastly, similar characteristics were found regarding to the distribution of ADRs by SOC and PT. Indeed, both general and local anaesthetics were associated with general and cutaneous disorders, with common ADRs that included a lack of efficacy, rash, and erythema. In addition, while general anaesthetics were associated with the occurrence of respiratory disorders (bronchospasms), local anaesthetics were associated with the occurrence of nervous disorders (paralysis).

Despite the encouraging data from the Italian spontaneous reporting system on anaesthetics, we should always keep in mind that ADRs still represent a common cause of illness and even death. Thus, the continuous monitoring of these drugs, together with the obligation to declare incidents and accidents occurring in operating rooms, is highly recommended, especially in operating rooms and medical clinics where the medical and paramedical staff should pay attention to the occurrence of any potential anaesthetic-induced ADR. In addition, the monitoring of the safety profile of these drugs is highly recommended, especially among frail populations, such as in paediatric populations for which drugs used to provide general anaesthesia have only been historically studied in adults. Thus, for this population safety and efficacy data mainly derive from real-life experience. In this context, pharmacovigilance activities have to be considered as a key component of an effective drug regulation system, clinical practice, and public health program in order to reduce harm to patients, improve public health, and reduce healthcare costs. Since underreporting still represents one of the major obstacles to the spontaneous reporting of ADRs, pharmacovigilance activities should be promoted via educational campaigns addressed to both healthcare professionals and patients. This will contribute to the encouragement of the rational and safe use of medicines.

## Figures and Tables

**Table 1 pharmaceuticals-14-01261-t001:** Adverse drug reactions associated with drugs used in general and local anaesthesia.

Drug(s)	Adverse Drug Reactions
Propofol	▪ Depression ▪ Hypotension ▪ Bradycardia and cardiac arrest ▪ Increase in triglycerides
Etomidate	▪ PONV
Ketamine	▪ Hypertension ▪ Hypersalivation ▪ Hallucinations
Inhalational anaesthetics (nitrous oxide, sevoflurane, desflurane, isofluran)	▪ Nephrotoxicity and hepatotoxicity ▪ Arrhythmias ▪ PONV ▪ Malignant hyperthermia and post-anaesthesia agitation
Thiopental	▪ Increase in blood pressure ▪ Nausea ▪ Risk of addiction, overdose, and death
Benzodiazepines	▪ Hypotension ▪ PONV * ▪ Thrombophlebitis, thrombosis, and pain at the injection ▪ Anterograde amnesia ▪ Drowsiness, ataxia, falls, and confusion (mainly in the elderly)
Synthetic opioids (sufentanil, remifentanil, fentanyl)	▪ Meiosis ▪ Respiratory depression ▪ Bradycardia ▪ PONV ▪ Urinary retention
Neuromuscular blocking drugs (succinylcholine, cisatracurium, rocuronium)	▪ Cardiovascular symptoms ▪ Cutaneous erythema
Local anaesthetics (lidocaine, mepivacaine, ropivacaine, bupivacaine, levobupivacaine, procaine)	▪ Application site reactions (pain, erythema, and oedema) ▪ Local anaesthetic systemic toxicity (rare)

* PONV: Postoperative nausea and vomiting.

**Table 2 pharmaceuticals-14-01261-t002:** Main characteristics of individual case safety reports which had general and local anaesthetic medicines as suspected drugs sent through the Campania region spontaneous reporting system, from January 2001 to September 2021.

Variable	Level	All ICSRs (*n* = 110)	ICSRs Reporting General Anaesthetic as Suspected Drugs (*n* = 50) *	ICSRs Reporting local Anaesthetic as Suspected Drugs (*n* = 60) **
Age, years	Median (IQR)	48.5 (30.75–64.25)	41 (16–59)	55 (39–69)
Sex	Female	51 (46.3)	22 (44)	29 (48)
	Male	56 (51)	26 (52)	30 (50)
	Missing	3 (2.7)	2 (4)	1 (2)
Seriousness	Hospitalisation or its prolongation	9 (8.2)	6 (12)	3 (5)
	Clinically relevant	17 (15.5)	6 (12)	11 (18)
	Life-threatening	14 (12.7)	11 (22)	3 (5)
	Not serious	64 (58.2)	25 (50)	39 (65)
	Not defined	6 (5.4)	2 (4)	4 (7)
Outcome	Unfavourable (resolved with sequelae or unchanged)	6 (5.4)	3 (6)	3 (5)
	Favourable (completely resolved or improvement)	78 (71)	37 (74)	41 (68)
	Not available	26 (23.6)	10 (20)	16 (27)

* General anaesthetics reported as suspected: cisatracurium (*n* = 1), fentanyl (*n* = 12), mivacurium (*n* = 2), propofol (*n* = 39), remifentanil (*n* = 1), rocuronium (*n* = 6), sevoflurane (*n* = 4), and sufentanil (*n* = 3). ** Local anaesthetics reported as suspected: articaine (*n* = 1), bupivacaine (*n* = 27), levobupivacaine (*n* = 6), lidocaine (*n* = 3), lidocaine/prilocaine (*n* = 2), mepivacaine (*n* = 7), prilocaine (*n* = 2), and ropivacaine (*n* = 14). The total number of suspected drugs exceeds the total number of ICSRs by general and local anaesthetics, since in a single ICSR more than one suspected drug can be reported.

**Table 3 pharmaceuticals-14-01261-t003:** Individual Case Safety Reports reporting general or local anaesthetic as suspected drugs and adverse drug reactions that were life-threating.

Case *n*.	Age	Sex	Outcome	ADR (s)	Suspected Drug (s)
1	60	F	UU	Quincke’s oedema, bronchospasm, rash	propofol, cisatracurium, sufentanil
2	63	M	NA	Hypotension, bradycardia	sevoflurane
3	48	F	FCR	Bradyarrhythmia	sufentanil
4	3	M	FCR	Cyanosis, Oxygen saturation abnormal	propofol
5	6	F	FCR	Cyanosis, Laryngospasm, Loss of consciousness, Tachycardia	propofol
6	4	M	FCR	Low oxygen saturation, tachycardia, temporomandibular joint dysfunction	propofol
7	44	F	FI	Rash	propofol
8	3	M	FCR	Bronchospasm, rash	propofol
9	7	F	FCR	Bronchospasm	propofol
10	2	M	FCR	Bronchospasm, rash, absence of pulse	propofol
11	54	F	FCR	Bronchospasm, diffuse erythema, oedema of the glottis, excessive bronchial secretion	propofol, fentanyl, rocuronium
12	67	M	URS	Hypotonia, flaccid paralysis	ropivacaine
13	49	M	FI	Accelerated hypertension, coma, respiratory failure, generalised convulsions, tachycardia	mepivacaine, levobupivacaine
14	26	F	FCR	Breathing difficulties, Sensation of crawling insects, hypophonesis	bupivacaine

FCR: Favourable–completely resolved; FI: favourable–improvement; NA: not available; URS: unfavourable–resolved with sequelae; UU: unfavourable–unchanged.

**Table 4 pharmaceuticals-14-01261-t004:** Distribution of adverse drug reactions related to general and local anaesthetic by System Organ Classes and Preferred Terms.

System Organ Class	General Anaesthetic (*n* = 86; 100%)	Local Anaesthetic (*n* = 90; 100%)
**Cardiac disorders *n*. (%)**	**6 (7)**	**3 (3)**
Bradyarrhythmia	1	-
Bradycardia	1	2
Paroxysmal atrial fibrillation	1	-
Tachycardia	3	1
**Gastrointestinal disorders**	**9 (10)**	**-**
Nausea	5	*-*
Vomiting	4	-
**General disorders and administration site conditions**	**11 (13)**	**52 (58)**
Face oedema	-	1
Lack of efficacy	10	42
Neck oedema	-	1
Pain	-	7
Pyrexia	1	1
**Investigations**	**8 (9)**	**1 (1)**
Absence of pulse	1	-
Hypophonesis	-	1
Hypotension	2	-
Increase in hepatic enzymes	3	-
Low oxygen saturation	2	-
**Musculoskeletal and connective tissue disorders**	**3 (3)**	**-**
Dysfunction of the temporomandibular joint	1	-
Muscle stiffness	1	-
Rhabdomyolysis	1	-
**Nervous system disorders**	**4 (5)**	**8 (9)**
Clonus	-	1
Coma	-	1
Dysesthesia	-	1
Hypotonia	-	1
Loss of consciousness	1	-
Paralysis	-	2
Psychomotor agitation	1	-
Seizure	-	1
Sensation of crawling insects	-	1
Tremors	2	-
**Renal and urinary disorders**	**3 (3)**	**-**
Abnormal urine colour	3	-
**Respiratory, thoracic** **, and mediastinal disorders**	**12 (14)**	**6 (7)**
Bronchospasm	5	1
Dyspnoea	1	-
Excessive bronchial secretion	1	-
Laryngospasm	1	-
Oedema of the glottis	1	-
Respiratory arrest	1	1
Respiratory failure	-	1
Shortness of breath	1	1
Throat constriction	1	2
**Skin and subcutaneous tissue disorders**	**19 (22)**	**10 (11)**
Angioedema	-	2
Desquamation	-	1
Erythema	4	1
Goose bumps	-	1
Petechiae	-	1
Pruritus	-	1
Quincke’s oedema	1	-
Rash	12	2
Redness	-	1
Sweating	2	-
**Vascular disorders n. (%)**	**5 (6)**	**5 (6)**
Cyanosis	3	1
Hypertension	-	1
Hypotension	1	3
Venous thrombosis	1	-
**Other SOCs**		
**Ear and labyrinth disorders**	**2 (2)**	**1 (1)**
**Eye disorders**	**1 (1)**	**-**
**Immune disorders**	**-**	**1 (1)**
**Injury, poisoning** **, and procedural complications**	**2 (2)**	**-**
**Investigations**	**-**	**1 (1)**
**Psychiatric disorders**	**1 (1)**	**-**
**Reproductive system and breast disorders**	**-**	**1 (1)**
**Surgical and medical procedures**	**-**	**1 (1)**

## Data Availability

The data presented in this study are available on request from the corresponding author. The data are not publicly available due to privacy and ethical reasons as established by the Italian Medicine Agency.

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
