# Peer review of "The Safety Profile of General and Local Anaesthetic Agents: Data Collected during 20 Years of Spontaneous Reporting Activities in the Campania Region (Southern Italy)"

_pharmaceuticals, 2021, doi:10.3390/ph14121261_

Round 1
Reviewer 1 Report
I think it is a good study using data accumulated over a long period of time.
Author Response
Dear reviewer, we'd like to thank you very much for your revision.
Reviewer 2 Report
Dear Authors,
Your manuscript entitled "The safety profile of general and local anaesthetic agents: data collected during 20 years of spontaneous reporting activities in the South of Italy" described the analyses of spontaneous delivered ICSRs during 20 years from a specific part of Italy regarding general or local anaesthetic and adverse effects.
- the title could present the specific site of the study
- in the Abstract, it seemed that the country missed
- please, consider putting together the first and second paragraphs
- in Introduction, a table with the adverse effects associated with the specific classes of anaesthetics would be welcome
- in line 215, who sent the 3% of the ICSRs?
- please, present a perspective to elevate the delivery of ICSRs, for instance, an educational campaign
- please, add a figure of the map where the study was perfomed in Italy.
Author Response
Dear reviewer, first of all we’d like to thank you for the time you spent in reviewing our manuscript. We took into account all your suggestions and we modified the manuscript accordingly.
Specifically, in the title we added “Campania Region” before “South of Italy” and we added “Southern Italy” in the abstract.
Please note that the second paragraph (“Methods”) was moved after the discussion as suggested by the Editor. Thus, we couldn’t put together the first and second paragraphs.
According to your suggestion, a new table reporting ADRs associated with anaesthetics was added. New data on the safety profile of these drugs were added in the manuscript as well (please see lines 82-87).
We specified that the 3% of the ICSRs were reported by patients/citizens (please see line 232-233) and we highlighted the importance of educational campaigns at line 264-368.
Lastly, due to copyright reasons we couldn’t use any figure reporting the Italian map. Therefore, we added a further table (as supplementary material) reporting all Italian regions and their population, where we highlighted the study site (Campania region).
Reviewer 3 Report
Thank you for permitting me to review this manuscript
The authors interrogated a data base (southern Italy ) in relation to individual case safety reports , with sponataneous reporting
I have some remarks
Please describe with a few sentence this database and how it works
who report? apparently it is patients and physician , theredore some inaccurracy could come from patients since they might not totally understand situations, this should be discussed
how was data extraction performed ? was it a database interrogation with sequential query language or other technique ? the incidence of missing data should be explained
what was the identity of the interrogators ?
what is the percentage of anesthesia performed in this region ?
May be a cross check with another database would have confirmed the incidence of the events declared , this also shoujd have been discussed
I have no doubt that all adverse events of any drugs should always be monitored and registered , however I do not think this kind of database which are partially biaised, and because of the spantaneous nature of data storage are the right system to monitor , on the contrary an obligation to declare accidents would ve much more appropriate
I cannot see any indication that dosage of anesthetics were appropriate and there was no anesthetics over dose, this should be discussed
Author Response
Dear reviewer, first of all we’d like to thank you for the time you spent in reviewing our manuscript. We took into account all your suggestions and we modified the manuscript accordingly.
Regarding to the description of the RNF, please see from line 304 to line 314 where we described in details what the RNF is and how it works. We also reported the methodology that we used for data extraction and the identity of interrogators (please see lines 315-320).
Regarding to the type of reporters, we specified that both HCPs and patients can fill the ICSR reporting form (line 308). Given the low number of ICSRs coming from patients, we provided our personal interpretation for this data (please see lines 235-240).
The importance to support our data with those deriving from other PV database has been discussed, as suggested (please see lines 277-280).
Unfortunately, we were not able to gather data on percentage of anesthesia performed in our region, instead we reported the most recent data on drugs’ utilization in Italy related to injective drugs (please see lines 69-73).
As suggested, the obligation to declare incidents and accidents was mentioned at lines 354-355.
Lastly, regarding to the incidence of missing data (including data related to anesthetics’ doses), we reported a personal interpretation of this phenomena when we discuss it in the section dedicated to study’s limitations (please see line 272-275).
Finally, please note that according to the second reviewer’ suggestions, 2 new tables were added (one reporting ADRs associated with anaesthetics and one reporting the main characteristics of Italian regions and their population, where we highlighted the study site). New data on the safety profile of these drugs were added in the manuscript as well (please see lines 82-87).
Round 2
Reviewer 3 Report
The authors have adequately improved the manuscript by answering most of reviewers query